# How Satisfied Are Women 6 Months after a Pessary Fitting for Pelvic Organ Prolapse?

**DOI:** 10.3390/jcm11195972

**Published:** 2022-10-10

**Authors:** Siegfried Nebel, Christian Creveuil, Michel Briex, Raffaèle Fauvet, Anne Villot, Anne-Cécile Pizzoferrato

**Affiliations:** 1Department of Obstetrics and Gynaecology, Caen University Hospital Center, 14000 Caen, France; 2Unit of Biostatistics and Clinical Research, Caen University Hospital, 14000 Caen, France; 3Department of Gynaecology, Obstetrics, Libourne Public Hospital Center, 33500 Libourne, France; 4Inserm U1086 “ANTICIPE”, Unité de Recherche Interdisciplinaire pour la Prévention et le Traitement des Cancers, 14000 Caen, France; 5Department of Obstetrics and Gynaecology, Cotentin Public Hospital Center, 50100 Cherbourg, France

**Keywords:** pelvic organ prolapse, pessary, quality of life, satisfaction

## Abstract

Background: The non-surgical solution for Pelvic Organ Prolapse (POP) typically consists of a pessary fitting. We aimed to assess patient satisfaction and symptom improvement 6 months after a pessary fitting and to identify risk factors associated with pessary failure. Methods: Six months after a pessary fitting, patient satisfaction was assessed by the PGII score; symptoms and quality of life were assessed using validated questionnaires (PFDI-20, ICIQ-SF, PISQ-12, USP, and PFIQ-7). Results: Of the 190 patients included in the study (mean age of 66.7 years), 141 (74%) and 113 (59%) completed the follow-up questionnaires at 1 and 6 months, respectively. Nearly all the women were menopausal (94.6%) and 45.2% declared being sexually active at inclusion. The satisfaction rate was 84.3% and 87.4% at 1 and 6 months, respectively. The global symptom score PFDI-20 had significantly improved at 6 months. A high body mass index (RR = 1.06, CI_95%_: [1.02–1.09]), as well as high PFDI-20 (1.05 [1.01–1.09]), PFIQ7 (1.04 [1.01, 1.08]), and PISQ12 scores at inclusion (0.75 [0.60, 0.93]), as well as higher GH and GH/TVL measurements (1.49 [1.25–1.78] and 1.39 [1.23–1.57], respectively) were associated with pessary failure. Conclusions: Pessary seems to be an effective treatment for POP with high patient satisfaction. Higher BMI, higher symptom scores, and greater genital hiatus measurements before insertion are risk factors for failure at 6 months.

## 1. Introduction

Symptomatic Pelvic Organ Prolapse (POP) in women can occur at any age and provokes symptoms that have a major impact on the patient’s quality of life. It is estimated prevalence varies widely in the literature from 4 to 12%, depending on the evaluation method (validated questionnaire or not) [1,2,3]. Women with POP should be followed by a multidisciplinary team and management should include both surgical and non-surgical techniques. The most common POP surgical procedures are anterior and posterior vaginal wall repair, laparoscopic sacropexy, or sacrospinous fixation by the vaginal route. Among the non-surgical solutions, the most common is pessary placement, which provides intravaginal support for the pelvic organs. Although the pessary is an effective treatment option for POP, it is often seen as a temporary solution whilst waiting for surgery, or as a therapeutic technique only when surgery is contraindicated. Few studies have assessed their effectiveness and satisfaction using validated questionnaires, and none have been carried out in a French population [4]. The main purpose of our study was to assess the satisfaction of patients 6 months after having been fitted with a pessary by means of a validated indicator (Patient Global Impression of Improvement, PGI-I) [5]. Secondary objectives were to evaluate improvement in symptom scores and to evaluate the risk factors associated with pessary failure.

## 2. Materials and Methods

### 2.1. Study Participants

We conducted a prospective observational cohort study in two French obstetrics and gynecology centers (Centre Hospitalier et Universitaire de Caen Normandie in Caen and Centre Hospitalier Public du Cotentin in Cherbourg). Patients were included from December 2018 to October 2021. The study was approved by a clinical and ethics board (Comité de Protection des Personnes Île de France, 2018-A00004-51) and complied under French law for personal data collection and analysis.

Inclusion criteria were women over 18 years of age who had been referred for symptomatic POP and who understood French. Women who were pregnant or breastfeeding, as well as women unable to fill in the forms, were excluded. Informed consent was obtained for all the women included.

### 2.2. Study Design

All the women were offered a pessary fitting as a first-line treatment. We have at our disposal the main sizes of the ring, cube, dish, and donut pessaries. The first option was the ring pessary, and the size was generally selected according to the distance between the vaginal fundus and the lower edge of the pubic symphysis as assessed by vaginal examination during the initial consultation. The stability of the pessary was checked by asking the patient to cough, strain, and walk around. If the ring pessary did not stay in place, a donut or cube pessary was tested. Dish pessaries were tested in women complaining of urinary incontinence and vaginal bulge.

At the first consultation, patients were asked to complete baseline questionnaires to assess their symptoms, their severity, and their impact on quality of life. These questionnaires included: the ICIQ-SF score (International Consultation Incontinence Questionnaire–Short Form) [6], the USP score (Urinary Symptoms Profile) [7] for urinary symptoms, the PFDI-20 score (Pelvic Floor Distress Inventory) [8] for symptoms associated with POP, the Kess and Wexner scores for anorectal symptoms [9,10], the PISQ-12 score (Pelvic organ prolapse/urinary Incontinence Sexual Questionnaire) [11] for assessment of sexual activity, and the PFIQ-7 score (Pelvic Floor Index Questionnaire) [8] for quality of life. The clinical examination consisted of an assessment of the type and stage of the POP using the standardized POP-Q system [12].

After the pessary fitting visit, the patients completed the same questionnaires at 1 and 6 months, as well as the PGI-I (Patient Global Impression of Improvement) score [5]. The PGI-I is a simple scale to assess the effectiveness of a treatment for a disease. In our study, a PGI-I score ≤ 2 was considered a positive response to pessary use. Patient satisfaction was evaluated using two additional questions: “Would you recommend this device to a friend?” and “For how long could you continue to use this device?”. Additional questions about pessary self-management (“Are you able to remove your pessary on your own?”), and side effects (vaginal discharge, a feeling of discomfort, and the need for manual positioning) were assessed at each time point.

Throughout the study period, patients who had not filled in the questionnaires could complete them at home and send them by post to their gynecologist.

The primary outcome of our study was patient satisfaction as assessed by a PGI-I score ≤ 2. Based on the literature and an expected percentage of satisfaction of 70%, we calculated that including at least 150 patients would allow us to estimate patients’ satisfaction with a 7.5% precision (half-width of the 95%CI).

Pessary failure was defined as the withdrawal of the pessary because of no clinical improvement, or at the patient’s request, at 6 months. Patients who wanted to try another pessary if the first did not stay in place could test as many as possible. In this case, we did not count it as a failure.

The women were encouraged to manage their pessaries alone at each consultation (every 2 weeks if necessary).

The MCID (Minimum clinical difference) of the PFDI-20 score was estimated based on the study by Wiegersma et al. [13] which found that a score improvement of 13.5 was clinically relevant.

### 2.3. Statistical Analyzes

Associations between risk factors and patient satisfaction based on the PGI-I index were assessed using univariate log-binomial regressions, allowing estimation of relative risks. Associations between risk factors and pessary failure at 6 months were first assessed using univariate log-binomial regressions. To account for interval censoring, hazards ratios in censored data analysis were also obtained using discrete Cox models. To compare changes in score distributions between 0, 1, and 6 months, we used mixed effects models. Statistical significance was defined as *p* < 0.05.

All statistical analyses were performed using IMB SPSS software (version 23, IBM, Armonk, NY, USA) and the R programming language.

## 3. Results

### 3.1. Patient Characteristics

During the study period, 200 women with symptomatic POP were offered a pessary fitting of whom 10 either refused the pessary or refused to participate in the study. The remaining 190 women (95%) were included in the study and asked to complete the inclusion questionnaires (Figure 1). Over the study period, 51 women experienced pessary failure, among whom 17 underwent subsequent surgery. The mean age of the patients was 66.7 (±9.8) years (Table 1).

More than half of the women (51.8%) had a professional activity at inclusion, and 62.8% declared having physical activity at least once a week. The patients were slightly overweight (mean BMI = 25.8 ± 4.5), mostly multiparous (82.4% had ≥2 vaginal deliveries), nearly all (94.6%) were menopausal, and 45.2% declared being sexually active at inclusion.

### 3.2. Symptoms Assessment and Patient Satisfaction

Among the 190 women who filled in the inclusion questionnaires, 141 (74%) completed the follow-up questionnaires at 1 month and 113 (59%) at 6 months. The reasons for non-completion were because of pessary failure or because the patient was lost to follow-up.

The main symptoms reported by the patients at inclusion were the presence of a vaginal bulge alone (55.7%) or associated with urinary leakage (36.9%).

As presented in Table 2, the most frequently affected compartment was the anterior vaginal wall (74.7%) and most patients had a POP-Q stage 2 or 3 prolapse (91.3%).

One hundred fifteen women (60.5%) were fitted with one pessary, 50 women (26.3%) tried a second one, and 16 (8.4%) a third or more. A total of 100 women (52.6%) had received associated local treatment.

The main outcome (PGI-I ≤ 2) was reached for 84.3% of responding patients at 1 month (102/121) and 87.4% at 6 months (97/111), showing a global satisfaction of the patients. Patients reported that the main benefits after fitting were the comfort and feeling of wellbeing provided by the pessary (87% and 85.4% at 1 and 6 months, respectively). When asked about the main disadvantage of the pessary, the most common response was ‘None’ (26.4% and 35.4% at 1 and 6 months, respectively). At 1 month, the patients complained about modified or increased vaginal discharge (16.5%), discomfort in the vagina (16.5%), the need for manipulation (23.1%), or sexual intercourse issues (2.2%). No severe adverse events occurred during the study period.

During follow-up, 76.6% of the patients at 1 month and 86.4% at 6 months were willing to continue the pessary for more than 5 years. More than 90% of the patients would have recommended the pessary to one of their friends at both 1 and 6 months, even though 69.3% at 1 month reported that they had not heard of the device before.

The global symptom score PFDI-20 significantly improved after the pessary fitting, with a mean score of 105.22 at inclusion, 56.91 at 1 month, and 57.48 at 6 months (*p* < 0.001 for global difference, *p* < 0.001 for M0–M1 difference, and *p* = 0.62 for M1–M6 difference). A significant improvement in the PFDI-20 score reaching the MCID or greater was observed for 73.5% and 72.8% of the women at 1 and 6 months. We found a significant improvement in the POPDI-6, UDI-6, and CRADI-8 subscores of the PFDI-20 with a similar profile. The USP subscore for overactive bladder (OAB) and low stream (LS) had also significantly improved (*p* = 0.004 and *p* < 0.001, respectively). There was no improvement in the USP subscore for stress urinary incontinence (SUI). Neither were any improvements observed for the Wexner and Kess scores (Table 3).

Only 69 patients answered the PISQ-12 questionnaire evaluating sexual discomfort at inclusion, 48 at 1 month, and 46 at 6 months, with no significant difference (*p* = 0.7). The global PFIQ7 score assessing quality of life had significantly improved at 1 month and 6 months (*p* < 0.001), as had the UIQ7, CRAIQ7, and POPIQ7 subscores (Table 3).

After 1 and 6 months, 62.8% and 67.8% of the patients, respectively, were able to remove the pessary alone (*p* = 0.4).

### 3.3. Factors Associated with Patient Satisfaction and Pessary Failure

Factors associated with greater patient satisfaction at 6 months were a history of hysterectomy (*p* < 0.001) and improvement in PFDI-20 at 1 month (*p* = 0.041). POP stage or age at inclusion were not found to be significantly associated with patient satisfaction at 6 months (Table 4).

In our population, 26.8% of the patients (51/190) were considered to have pessary failure at 6 months, either because the pessary fell out or because it was poorly tolerated. Failure of the pessary at 6 months was significantly associated with BMI at inclusion (RR = 1.06 for each increase of one point of the BMI, IC_95%_: [1.02–1.09], *p* < 0.01), with the genital hiatus (GH) measurement alone (RR = 1.49 for each increase of 10 mm, CI_95%_: [1.25–1.78], *p* < 0.001), and with the genital hiatus to total vaginal length ratio (GH/TVL, RR = 1.39 for each increase of 0.1, CI_95%_: [1.23–1.57], *p* < 0.001). Failure was also associated with the PFDI-20 score at inclusion (RR = 1.05 for each increase of 10 units of the score, CI_95%_: [1.01–1.09], *p* = 0.02), with the PFIQ7 score at inclusion (RR = 1.04 for each increase of 10 units of the score, CI_95%_ = [1.01, 1.08], *p* = 0.01), and with the PISQ12 score at inclusion (RR = 0.75 for each increase of 5 units of the score, CI_95%_ = [0.60, 0.93], *p* = 0.01). Parity, age at inclusion, menopausal status, smoking, surgical history, and POP stage were not found to be significantly associated with failure at 6 months (Table 5). In our population, only six women declared de novo SUI (5.3%) six months after the pessary fitting. At six months, de novo UI was not associated with pessary failure.

In censored data analysis, hazard ratios were similar to the relative risks described above for all factors, including GH and GH/TVL measurements, but not for the PFIQ-7, which lost statistical significance (HR = 1.04, IC_95%_: [1.00–1.09], *p* = 0.053).

## 4. Discussion

Our study showed that pessary placement in women presenting with POP is well accepted with a high satisfaction rate at 6 months in the women continuing to use a pessary: 87.4% of women reported being improved or very improved, and more than 90% of them would recommend the pessary to one of their friends. Factors associated with greater patient satisfaction at 6 months were a history of hysterectomy and improvement in PFDI-20 at 1 month. Most of the symptom scores had significantly improved as early as 1 month after the pessary fitting, and the improvement was maintained at 6 months of follow-up. Pessary failure was associated with a high BMI, a higher PFDI-20 score, a lower PISQ-12 score, and greater GH and GH/TVL measurements.

By the end of follow-up, 68% had dropped out of the study, which is comparable to what is found in the literature. Our population was also of a similar age as populations in other published studies.

We found a very high pessary acceptance rate at inclusion (95% of women accepted a pessary fitting at inclusion), even though more than half of the women reported that they had not heard of the device before. This would indicate that pessary usage is very likely to be accepted by patients as a first-line treatment. Other authors have reported a low acceptance rate of pessary fittings, especially among younger women, women with a higher POP-Q stage, lower education level, and those who had previously been consulted for POP [14,15].

Our patient satisfaction rate is similar to that reported in the literature: two studies assessing pessary satisfaction with the PGI-I found comparable satisfaction rates of 83% at 1 year and 86.9% at 6 months [16,17]. However, the factors associated with patient satisfaction were not analyzed in these studies. In our study, a greater improvement in PFDI-20 scores (before insertion vs. at 1 month) was significantly associated with greater patient satisfaction. One should take into account that we had more women with high-stage POP (i.e., stages 3–4) than with low-stage POP in our study. While it might be thought that this would result in an inflated satisfaction rate, we did not find any statistically significant association between POP stage and patient satisfaction at 6 months. This has been described in previous observational studies [18,19] and might support the use of pessaries in patients with high-stage POP.

Furthermore, it is interesting to note that satisfaction seemed to improve over time, which probably indicates that women need time to gain confidence and learn how to use the device. The greatest improvement in the patients’ symptoms occurred early after the fitting, since significance was reached by the first month of comparison in our study, and not between the first and sixth month. Therefore, it can be assumed that the pessary is effective early on and that effectiveness is maintained over time.

In the published literature, several different symptom scales have been used to assess pessary satisfaction, but not all have been validated for studies. The most commonly used validated questionnaires on pelvic symptoms are the PFDI-20 and PFIQ7 scores, for which we found similar improvements to those of most authors. For example, in a study of 84 women in 2019, Mao et al. found that the PFDI-20 score dropped from a mean of 98.4 to 46.6, and that the PFIQ-7 score dropped from a mean of 77.3 to 13.2 after 23 months of follow-up [20].

The improvement in urinary symptoms, as assessed mainly through the USP score, was probably mainly due to improvements in the symptoms of OAB and LS complaints. This finding is consistent with the rest of the literature. For example, Fernando et al. [21] studied 203 women referred for symptomatic POP with a mean age of 69 years. Their primary endpoint was the change in symptoms 4 months after a pessary fitting, measured with the Sheffield questionnaire. Thirty-nine women (40.2%, *p* = 0.001) declared an improvement in bladder emptying, and 37 (38%, *p* = 0.001) declared an improvement in urinary urgency.

The literature also reports a varying range of patients experiencing persistence of OAB even after 2 to 4 months of pessary use [21,22,23]. In our data, we found an early improvement of the USP subscores for OAB and LS at 1 month compared with the baseline, with no subsequent statistical difference found between the first and sixth month. Results for the PFIQ-7 urinary subscore, the UIQ-7, also indicate that patients felt less impaired by their urinary symptoms after 1 month of treatment, with no difference at 6 months. Furthermore, our population had rather low baseline scores (mean USP-OAB at baseline = 6.45/21, mean USP-LS at baseline = 1.26/9), indicating that their symptoms were already mild, and any change would consequently be more difficult to assess. Overall, this suggests that the pessary is highly effective for rapid urinary symptom relief, even in mildly symptomatic patients. Improvement in OAB symptoms may be due to an improvement in voiding symptoms with pessary use by restoration of the bladder and urethra anatomy. It is therefore possible that POP promotes secondary detrusor overactivity that was rectified by the pessary.

The absence of a difference between 1 and 6 months suggests that no major secondary worsening of urinary symptoms occurs during the treatment. No difference was observed for the USP-SUI subscore, and urinary leakage at 1 month was not associated with lower patient satisfaction. Although some authors find a de novo UI rate of up to 23% [22], the association between de novo UI and patient dissatisfaction is inconsistent in the literature [18,22]. Nevertheless, informing patients about the risk of de novo UI is essential and likely to improve patient satisfaction.

Sexual symptoms were not found to be significantly improved by the use of a pessary in our study, but only around 36% of the patients completed this questionnaire at inclusion. In 2009, Kuhn et al. [24] assessed sexual function in a prospective cohort of 31 women with a mean age of 70 years, before and 3 months after the use of a cubic pessary using the Female Sexual Function Index (FSFI). The score for sexual desire improved significantly from 6 to 14, and the score for satisfaction improved significantly from 1 to 4.

In our population, the main complaints related to pessary use were the need for manual positioning, the occurrence of vaginal discharge, and a feeling of discomfort, which is consistent with the literature [25].

Risk factors for pessary failure were concordant with the literature. In 2007, Komesu et al. [26] also found that a high PFDI-20 score was associated with a higher risk of failure. Interestingly, we found that a high PISQ-12 score was associated with a much-reduced risk for the pessary treatment to fail (a 5-point increase reduced the risk by 25%). However, as stated earlier, our PISQ-12 data was probably flawed because of a poor response rate. The literature is inconsistent about the risk factors for pessary failure. Clemons et al. found that older age was a factor for pessary success [22], whereas Yang et al. found that a posterior prolapse was more likely to result in failure [27]. Several studies have also shown that a short vaginal length and a large GH increase the risk of failure [20,22,28].

A recurrent complaint about pessary use is the need for regular medical follow-up, especially to remove and re-position it [29]. One solution to address this issue would be to teach women how to position the pessary themselves. There are however few robust data to scientifically support this practice [30]. In our study, 65% of the patients declared that they knew how to remove their pessary on their own. Although no statistical difference was found when comparing the satisfaction score or the number of failures, it would seem important to continue to explore and promote self-management of the pessary.

Limitations of our study include the number of patients lost to follow-up or with missing data. We were unable to evaluate why some questionnaires were not completed. In addition, patient satisfaction is probably overestimated because women who had a pessary failure did not respond to questionnaires. Moreover, the interpretation of the results of this study should be moderated due to the lack of a control group.

Future studies should include a longer follow-up to detect factors associated with pessary failure and help healthcare professionals manage their patients who suffer from POP.

## 5. Conclusions

The pessary is an effective treatment option for the management of POP since it improves many of the associated symptoms. It is well accepted, rapidly effective, and its effects seem to last over time.

## Figures and Tables

**Figure 1 jcm-11-05972-f001:**
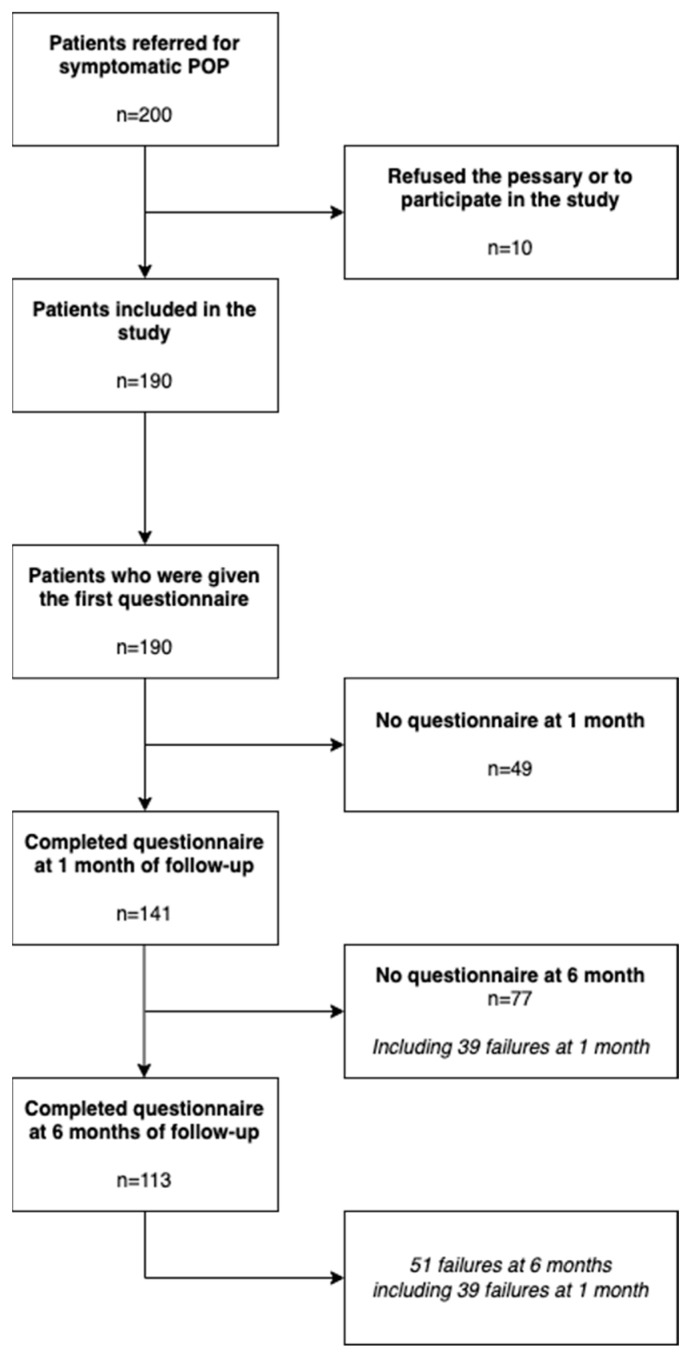
Flow chart of the patients recruited in our study (POP: Pelvic Organ Prolapse).

**Table 1 jcm-11-05972-t001:** Baseline characteristics of the population at inclusion.

		*n* (%)
Age (Years) [m (sd)]		66.7 (9.8)
BMI ^a^ (kg/m^2^) [m (sd)]		25.8 (4.5)
Parity [Median (Quartiles)]		2 (2; 3)
Number of vaginal deliveries	0 1 2 ≥3	7 (3.7%) 26 (13.9%) 68 (36.4%) 86 (46.0%)
Graduate degree or more	No Yes	91 (54.5%) 76 (45.5%)
Menopausal status	No Yes, without HRT ^b^ Yes, with HRT ^b^	10 (5.4%) 133 (71.9%) 42 (22.7%)
Smoking	No Yes	170 (93.9%) 11 (6.1%)
Diabetes	No Yes	167 (89.3%) 20 (10.7%)
Hypertension	No Yes	119 (64.0%) 67 (36.0%)
Neurological disease	No Yes	179 (96.2%) 7 (3.8%)
History of pelvic surgery	No Yes	94 (50.5%) 92 (49.5%)
Hysterectomy	No Yes	156 (84.8%) 28 (15.2%)
Sexually active	No Yes	86 (54.8%) 71 (45.2%)

^a^ BMI: Body Mass Index; ^b^ HRT: Hormone Replacement Therapy.

**Table 2 jcm-11-05972-t002:** Clinical evaluation and symptoms at inclusion.

		*n* (%)
Main complaint reported by women	Vaginal bulge and UI	65 (36.9%)
	Vaginal bulge alone	98 (55.7%)
	Urinary incontinence	8 (4.5%)
	Others	5 (2.8%)
EQ-5D ^a^ (mean value at inclusion, *n* = 150)	Mobility	1.193
	Self-care	1.093
	Usual activities	1.280
	Pain/discomfort	1.767
	Anxiety/depression	1.729
Discomfort scale [m (sd)]		7.18 (2.37)
POP ^b^ staging (using POP-Q ^c^ classification)	1	5 (2.7%)
	2	99 (53.5%)
	3	70 (37.8%)
	4	11 (5.9%)
Cystocele	0	4 (2.1%)
	1	25 (13.4%)
	2	101 (54.0%)
	3	52 (27.8%)
	4	5 (2.7%)
Hysteroptosis	0	56 (30.6%)
	1	53 (29.0%)
	2	36 (19.7%)
	3	26 (14.2%)
	4	12 (6.6%)
Rectocele	0	79 (43.2%)
	1	51 (27.9%)
	2	41 (22.4%)
	3	11 (6.0%)
	4	1 (0.5%)

^a^ EQ-5D: EuroQol–5 Dimension questionnaire; ^b^ POP: Pelvic Organ Prolapse; ^c^ POP-Q: Pelvic Organ Prolapse Quantification.

**Table 3 jcm-11-05972-t003:** Changes in women’s symptoms and quality of life using mixed effects models.

	M0 Mean (±SD)	M1 Mean (±SD)	M6 Mean (±SD)	Global *p*	*p*	*p*
					M0–M1	M1–M6
POPDI-6	45.62 (±23.29)	18.10 (±21.82)	18.23 (±20.63)	<0.001	<0.001	0.65
CRADI-8	23.98 (±20.52)	18.08 (±18.56)	17.15 (±16.61)	<0.001	<0.001	0.88
UDI-6	36.38 (±24.79)	22.19 (±23.77)	21.97 (±21.46)	<0.001	<0.001	0.86
PFDI-20	105.22 (±53.35)	56.91 (±50.25)	57.48 (±46.47)	<0.001	<0.001	0.62
ICIQ-SF	5.96 (±5.64)	5.04 (±05.30)	4.24 (±04.42)	0.0082	0.017	0.59
USP SUI	2.09 (±2.67)	1.86 (±02.79)	1.78 (±02.42)	0.26		
USP OAB	6.45 (±4.12)	5.49 (±04.45)	4.92 (±03.88)	0.0021	0.004	0.74
USP LS	1.26 (±1.35)	0.71 (±01.43)	0.81 (±01.37)	<0.001	<0.001	0.84
Wexner	3.60 (±4.37)	3.13 (±04.31)	3.32 (±04.10)	0.44		
Kess	10.05 (±8.07)	8.91 (±08.27)	8.19 (±07.34)	0.4		
PISQ12	32.30 (±7.03)	33.10 (±06.95)	34.24 (±06.50)	0.7		
UIQ7	25.60 (±27.50)	12.70 (±19.35)	10.46 (±16.80)	<0.001	<0.001	0.86
CRAIQ7	14.68 (±23.32)	7.33 (±15.10)	7.40 (±16.12)	0.0051	0.0036	0.92
POPIQ7	22.35 (±28.40)	8.35 (±15.73)	7.28 (±15.95)	<0.001	<0.001	0.94
PFIQ7	64.01 (±59.27)	28.28 (±40.64)	25.51 (±43.14)	<0.001	<0.001	0.92
EQ-5D	6.57 (±01.93)	18.93 (±25.86)	18.24 (±25.45)	<0.001	<0.001	0.94

POPDI-6: Pelvic Organ Prolapse Distress Inventory; CRADI-8: Colorectal-Anal Distress Inventory; UDI-6: Urinary Distress Inventory; PFDI-20: Pelvic Floor Disorder Inventory; ICIQ-SF: International Consultation on Incontinence Questionnaire–Short Form; USP: Urinary Symptom Profile; SUI: Stress Urinary Incontinence; OAB: Overactive Bladder; LS: Low Stream; PISQ 12: Pelvic organ prolapse urinary Incontinence Sexual Questionnaire; UIQ7: Urinary Impact Questionnaire 7; CRAIQ-7: Colorectal-Anal Impact Questionnaire; POPIQ-7: Pelvic Organ Prolapse–Impact Questionnaire; PFIQ-7: Pelvic Floor Impact Questionnaire; and EQ-5D: EuroQol–5 Dimension questionnaire.

**Table 4 jcm-11-05972-t004:** Factors associated with patient satisfaction at 6 months (using log-binomial regression).

Risk Factors	Patient Satisfaction at 6 Months		
	*n* (%)	RR (CI 95%)	*p*
Age (increase of 5 years)		1.04 (1.00; 1.08)	0.080
BMI (increase of 1 unit)		0.98 (0.96; 1.00)	0.10
Parity			
1	12/13 (92.3%)	1	
2	41/47 (87.2%)	0.95 (0.78; 1.14)	
≥3	43/50 (86.0%)	0.93 (0.77; 1.13)	0.77
Menopausal status			
No	3/5 (60.0%)	1	
Yes	93/105 (88.6%)	1.48 (0.72; 3.03)	0.29
Smoking			
No	87/99 (87.9%)	1	
Yes	8/9 (88.9%)	1.01 (0.79; 1.29)	0.93
History of pelvic surgery		1	
No	53/59 (89.8%)		
Yes	43/50 (86.0%)	0.96 (0.83; 1.10)	0.54
Hysterectomy			
No	83/97 (85.6%)	1	
Yes	11/11 (100%)	1.17 (1.08; 1.27)	<0.001
POP stage			
1–2	55/65 (84.6%)	1	
3–4	40/44 (90.9%)	1.07 (0.93; 1.24)	0.31
Manual repositioning			
No	57/64 (89.1%)	1	
Yes	30/36 (83.3%)	0.94 (0.79; 1.11)	0.44
Global discomfort because of POP (increase of 1 unit)		1.00 (0.97; 1.04)	0.84
PFDI-20 score before pessary (increase of 10 units)		0.99 (0.97; 1.00)	0.086
PFIQ-7 score before pessary (increase of 10 units)		1.01 (1.00; 1.02)	0.26
PISQ-12 score at inclusion (increase of 10 units)		0.93 (0.83; 1.05)	0.25
Change in PFDI-20 M1 vs. before pessary (increase of 10 units)		0.98 (0.96; 0.999)	0.041
Change in PFIQ-7 M1 vs. before pessary (increase of 10 units)		0.99 (0.97; 1.00)	0.088
Urinary leakage before pessary			
No	26/30 (86.7%)	1	
Yes	52/61 (85.2%)	0.98 (0.83; 1.17)	0.85
Urinary leakage at M1			
No	31/33 (93.9%)	1	
Yes	46/55 (83.6%)	0.89 (0.77; 1.03)	0.12

RR: Relative Risk; CI95%: 95% Confidence Interval; BMI: Body Mass Index; POP: Pelvic Organ Prolapse; PFDI-20: Pelvic Floor Disorder Inventory; PFIQ-7: Pelvic Floor Impact Questionnaire; and PISQ-12: Pelvic organ prolapse urinary Incontinence Sexual Questionnaire; M1: at one Month.

**Table 5 jcm-11-05972-t005:** Factors associated with pessary failure at six months (using log-binomial regression).

Risk Factors	Pessary Failure		
	*n* (%)	RR (CI 95%)	*p*
Age (increase of 5 years)	0.92 (0.84; 1.01)		0.072
BMI (increase of 1 unit)	1.06 (1.02; 1.09)		0.0022
Parity			
1	3/16 (18.8%)	1	
2	17/64 (26.6%)	1.42 (0.47; 4.25)	
≥3	31/78 (39.7%)	2.12 (0.74; 6.09)	0.14
Menopausal status			
No	3/8 (37.5%)	1	
Yes	47/149 (31.5%)	0.84 (0.33; 2.12)	0.71
Smoking			
No	49/146 (33.6%)	1	
Yes	0/9 (0.0%)	NA	0.058
History of pelvic surgery			
No	21/76 (27.6%)	1	
Yes	30/81 (37.0%)	1.34 (0.84; 2.13)	0.21
Hysterectomy			
No	40/129 (31.0%)	1	
Yes	11/26 (42.3%)	1.36 (0.81; 2.29)	0.24
POP stage			
1–2	24/88 (27.3%)	1	
3–4	26/67 (38.8%)	1.42 (0.90; 2.24)	0.13
Manual repositioning			
No	35/97 (36.1%)	1	
Yes	14/47 (29.8%)	0.83 (0.49; 1.38)	0.46
GH measure (increase of 10 mm)	1.49 (1.25; 1.78)		<0.001
GH/TVL ratio (increase of 0.10)	1.39 (1.23; 1.57)		<0.001
Discomfort because of POP (increase of one unit)	1.02 (0.92; 1.14)		0.69
PFDI-20 score (increase of 10 units)	1.05 (1.01; 1.09)		0.021
PFIQ-7 score (increase of 10 units)	1.04 (1.01; 1.08)		0.012
PISQ-12 score (increase of 5 units)	0.75 (0.60; 0.93)		0.010
Kess score (increase of 5 units)	1.07 (0.95; 1.21)		0.28
Urinary incontinence			
No	13/39 (33.3%)	1	
Yes	36/96 (37.5%)	1.12 (0.67; 1.88)	0.65

RR: Relative Risk; CI95%: 95% Confidence Interval; BMI: Body Mass Index; GH: Genital Hiatus; TVL: Total Vaginal Length; POP: Pelvic Organ Prolapse; PFDI-20: Pelvic Floor Disorder Inventory; PFIQ-7: Pelvic Floor Impact Questionnaire; and PISQ-12: Pelvic organ prolapse urinary Incontinence Sexual Questionnaire; GH: Genital Hiatus; TVL: Total Vaginal Length.

## Data Availability

Not applicable.

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
