# Peer review of "How Satisfied Are Women 6 Months after a Pessary Fitting for Pelvic Organ Prolapse?"

_jcm, 2022, doi:10.3390/jcm11195972_

Round 1
Reviewer 1 Report
General comments:
This prospective observational cohort study evaluated the satisfaction of pessary treatment in patients with POP.
The topic is relevant to the journal and relevant in the field of pelvic organ prolapse.
The methods are mostly clear and appropriate.
The outcome measures were evaluated with validated tools.
However, the study has some limitations, including the fact that a control group was lacking.
Specific comments:
Line 40: I would not state that the most common surgical solution for POP is laparoscopic sacropexy, as the choice of a surgical procedure varies very much worldwide. Instead I would suggest to name the most common methods of POP surgery.
How was the choice of pessary type made? Was sexual activity assessed, and did it influence the choice of pessary type.
It would have been interesting to know how many women were sexually active and to include this information into the baseline characteristics.
The sample size seems sufficient, however there is no report of a power analysis. Has this been performed?
It is reported that 26.8% (51/190) of patients had pessary failure at 6 months. Then again it is reported that 58 women experienced pessary failure over the study period. Please elaborate on how many patients actually had pessary failure by the first month and how many by the 6th month?
A flow chart of the study participants would assist additional clarity.
The majority of women were diagnosed with a prolapse stage of 2, which would explain the high satisfaction rate after pessary fitting, a conservative therapy enabling the treatment of minor prolapse stages rather than higher stages. Interestingly, the prolapse stage was not found to be associated with patient satisfaction at 6 months. This should be discussed further.
Please report if any adverse events occurred.
Please change the French to English in Table 4.
Interestingly, no differences were observed for the USP-SUI subscore, meaning no occult stress urinary incontinence was detected after pessary insertion. The role of pessaries in SUI could be discussed further.
Overall, I consider this an interesting and well conducted trial. However, the interpretation of the results needs to be treated with caution, as the drop-out rate was high, and women with pessary failure did not respond to the questionnaires. This is recognized by the authors themselves.
Despite these points of criticism, I consider this manuscript to be well written and a useful source of information for this interesting topic.
Reviewer 2 Report
Dear Authors,
I wish to congratulate you for this interesting study.
Please see my remarks below.
Abstract
The abstract doesn’t include any information regarding the percentage of sexually active / inactive patients.
Introduction
· Line 35: “Pelvic Organ Prolapse (POP) in women is a major public health concern”- this remark is an overstatement. Symptomatic prolapse is a major QoL issue in women would be a more accurate approach.
· Lines 35-37: “Its estimated prevalence varies widely – from 2.9% to 97.7% – depending on how the diagnosis is made 36 and whether the Pelvic Organ Prolapse Quantification method is applied or not”- please focus on the prevalence of symptomatic POP. You wouldn’t offer a pessary top asymptomatic women with POP.
· Line 40: “The most commonly performed surgical solution is laparoscopic sacropexy.” This sentence is both inaccurate and unreferenced
· Line 44-45: “However, recent guidelines of the French Health Ministry…”- wouldn’t you like to quote international guidelines like the IUGA/ICS consensus on POP treatment?
· Line 50: “and patient knowledge”- what do you mean? The meaning maybe is lost in translation and is not clear.
· In general, the introduction should focus a little more on what is known and what is yet unknown on patient satisfaction following pessary treatment. The introduction should present the rationale for your study objective: in what does this study differ from previous ones on the topic? What do you aim to add to previous knowledge?
Materials and methods
· Line 62-65: “All the women were offered a pessary fitting as a first-line treatment. The first option was the ring pessary, followed by donut or cube pessaries if the ring pessary did not stay in place. Dish pessaries were tested in women complaining of urinary incontinence and vaginal bulge”- did you use a “pessary fitting set”? Please describe the fitting process. Did you just measure and send the patients to buy the pessary? Or did you insert a pessary and let them ambulate/strain/sit on a toilet stool?
· Line 65-67: “The size of the pessary was generally selected according to the distance between the vaginal fundus and the lower edge of the pubic symphysis as assessed by vaginal examination during the initial consultation” – is there a reference for this measuring technique?
· Lines 68-75: you gave reference for some French versions of the questionnaires and for some other the reference is the original English validation. Were all questionnaires validated in French? Please specify. Moreover, you should explain why you chose to administer so many questionnaires some of which assess similar symptomatology.
· Line 71-72: “the USP score (Urinary Symptoms Profile) for urinary symptoms, the PFDI-20 score (Pelvic Floor Distress Inventory)” – please explain the reason to include both of those questioners in the study, what is the added value of USP questionnaire on the UDI-6 questionnaire (which is a part of PFDI – 20)
· Line 89-90: “Pessary failure was defined as the withdrawal of the pessary because of no clinical improvement or on the patient’s request, at 6 months”- how did you consider cases in which the pessary “fell out”? As a failure? Did you carry out a new fitting session of withdraw from the study? You should better define “pessary failure”.
· Lines 90-96: the description of statistical methods should appear as a new paragraph.
Results:
· Line 99: “During the study period, 200 women with symptomatic POP were offered a pessary fitting.” – there is no reference to the power calculation of the study, please address in the materials and methods section.
· Line 121-122: “The main outcome (PGI-I ≤ 2) was reached for 84.3% of responding patients at 1 month (102/121) and 87.4% at 6 month (97/111)” and line 154-155: “In our population, 26.8% of the patients (51/190) were considered to have pessary failure at 6 months, either because the pessary fell out or because it was poorly tolerated”: these two sentences are contradictory and confusing. How can the satisfaction rate be so high while ¼ of patients reported pessary failure at 6 months?
· Results on the PFDI-20 and its domains should calculate the minimal clinically important difference and report the percentage of women who reached such improvement.
· Table 1-5: None of the tables includes any data regarding sexual activity of the patients which is crucial when addressing women’s satisfaction.
· Table 1 and 2: The graphics can be improved, splitting the first columns into 2 separate ones.
· Table 4: some of the text is in French
· You should provide data on recommendations for pessary management: did the women self-manage? Or did they return to the clinic for periodical pessary care? Did the ability to self-manage impact pessary treatment success?
Discussion:
· Line 190: “Our satisfaction rate”- add the word “patients”
· Line 196: “to gain confidence” is better than “to trust”
· Line 208-213: “We did not find a significant improvement in the Wexner and Kess anorectal and digestive scores after pessary use. However, the scores were already low at baseline which suggests that a recruitment bias may explain this apparent lack of effectiveness. The CRADI-8 and CRAIQ-7 scores, components of PFDI-20 and PFIQ-7 questionnaires, did however statistically improve with the use of a pessary – a finding which is inconsistently reported in the literature and associated with poor clinical significance” –If you wish to discuss the fecal incontinence/constipation data, you should explain the rationale of how a pessary may impact colorectal symptoms.
· Line 214 - 215: “The improvement in urinary symptoms as assessed mainly through the USP score, were probably mainly due to improvements in the symptoms of OAB and LS complaints.” – please explain how in your opinion pessary improves OAB symptoms?
· Line 222-234: what about occult SUI becoming overt? This is important information to address and discuss
· Line 235: “Sexual symptoms were not found to be significantly improved by the use of a pessary” - here you address the sexual aspect for the first time. As mentioned above, it is an important information that was not conveyed anywhere before, despite its significance on the discussed topic. Please refer to the sexual activity and satisfaction at the very beginning of your manuscript.
· Line 236: “only around 36% of the patients completed this questionnaire“ - you mention that only 36% of the participants answered the sexual function questionnaire. Why do you think majority of patients did not fill it? Is it because it is suitable only for heterosexual coitally active partnered women? You should explain.
· Lines 241-243: “In our population the main complaints related to pessary use was the need for manual positioning, the occurrence of vaginal discharge, and a feeling of discomfort, which is consistent with the literature”- this is the first time you describe collecting such data- this should be presented in the M&M section and in the results, and then explained in the discussion.
· Line 244-246: “Pessary use failure was associated with a higher BMI (each point of increase in BMI increased the risk of failure at 6 months by 6%), PFDI-20 (a 10-point increase resulted in a 5% risk increase), PFIQ-7 (a 10-point increase resulted in a 4% increase), and greater GH 246 and GH/TVL measurements”. – this is just repeating the results, instead you should explain the meaning of your findings including in light of previous literature.
· Lines 248-252: a discussion of PISQ-12 and its low response rate is pointless if we don’t know how many women were sexually active and had a partner. Even with a low response rate’ you should make the effort to discuss a possible link between better sexual function and pessary failure. And compare to previous data in the literature.
· Line 260-261: we read this information here for the first time. Again, please address this in M&M and results. Did you ask all the patients? Was this data collected as part of a self-administered questionnaire? Or retrieved from clinical files? Or phone interview?
· Line 265: “Limits of our study “ – in general the manuscript should be reviewed by native English speaker. The appropriate expression is: Limitations of our study
To conclude, I believe the manuscript needs major revisions.
I also recommend language and style editing by a native English speaker.
Kind regards!
